# BRECQ: PUSHING THE LIMIT OF POST-TRAINING QUANTIZATION BY BLOCK RECONSTRUCTION

**Yuhang Li[12]\*, Ruihao Gong[2]\*, Xu Tan[2], Yang Yang[2], Peng Hu[2],**
**Qi Zhang[2], Fengwei Yu[2], Wei Wang, Shi Gu[1]✉**
[1]University of Electronic Science and Technology of China, [2]SenseTime Research
`liyuhang699@gmail.com, gongruihao@sensetime.com, gus@uestc.edu.cn`

## ABSTRACT

We study the challenging task of neural network quantization without end-to-end retraining, called Post-training Quantization (PTQ). PTQ usually requires a small subset of training data but produces less powerful quantized models than Quantization-Aware Training (QAT). In this work, we propose a novel PTQ framework, dubbed BRECQ, which pushes the limits of bitwidth in PTQ down to INT2 for the first time. BRECQ leverages the basic building blocks in neural networks and reconstructs them one-by-one. In a comprehensive theoretical study of the second-order error, we show that BRECQ achieves a good balance between cross-layer dependency and generalization error. To further employ the power of quantization, the mixed precision technique is incorporated in our framework by approximating the inter-layer and intra-layer sensitivity. Extensive experiments on various handcrafted and searched neural architectures are conducted for both image classification and object detection tasks. And for the first time we prove that, without bells and whistles, PTQ can attain 4-bit ResNet and MobileNetV2 comparable with QAT and enjoy 240× faster production of quantized models. Codes are available at `https://github.com/yhhhli/BRECQ`.

## 1 INTRODUCTION

The past decade has witnessed the rapid development of deep learning in many tasks, such as computer vision, autonomous driving, etc. However, the issue of huge computation cost and memory footprint requirements in deep learning has received considerable attention. Some works such as neural architecture search (Zoph & Le, 2016) try to design and search a tiny network, while others, like quantization (Hubara et al., 2017), and network pruning (Han et al., 2015) are designed to compress and accelerate off-the-shelf well-trained redundant networks.

Many popular quantization and network pruning methods follow a simple pipeline: training the original model and then finetune the quantized/pruned model. However, this pipeline requires a full training dataset and many computation resources to perform end-to-end backpropagation, which will greatly delay the production cycle of compressed models. Besides, not all training data are always ready-to-use considering the privacy problem. Therefore, there is more demand in industry for quantizing the neural networks without retraining, which is called Post-training Quantization. Although PTQ is fast and light, it suffers from severe accuracy degeneration when the quantization precision is low. For example, DFQ (Nagel et al., 2019) can quantize ResNet-18 to 8-bit without accuracy loss (69.7% top-1 accuracy) but in 4-bit quantization, it can only achieve 39% top-1 accuracy. The primary reason is the approximation in the parameter space is not equivalent to the approximation in model space thus we cannot assure the optimal minimization on the final task loss.

Recent works like (Nagel et al., 2020) recognized the problem and analyzed the loss degradation by Taylor series expansion. Analysis of the second-order error term indicates we can reconstruct each layer output to approximate the task loss degeneration. However, their work cannot further quantize the weights into INT2 because the cross-layer dependency in the Hessian matrix cannot be ignored when the perturbation on weight is not small enough. In this work, we analyze the second-order

---

\*Equal contribution, ✉ Corresponding author.

error based on the Gauss-Newton matrix. We show that the second-order error can be transformed into network final outputs but suffer from bad generalization. To achieve the best tradeoff, we adopt an intermediate choice, block reconstruction. In addition, our contributions are threefold:

1. Based on the second-order analysis, we define a set of reconstruction units and show that block reconstruction is the best choice with the support from theoretical and empirical evidence. We also use Fisher Information Matrix to assign each pre-activation with an importance measure during reconstruction.
2. We incorporate genetic algorithm and the well-defined intra-block sensitivity measure to generate latency and size guaranteed mixed precision quantized neural networks, which fulfills a general improvement on both specialized hardware (FPGA) and general hardware (ARM CPU).
3. We conduct extensive experiments to verify our proposed methods. We find that our method is applicable to a large variety of tasks and models. Moreover, we show that post-training quantization can quantize weights to INT2 without significant accuracy loss for the first time.

## 2 PRELIMINARIES

**Notations** Vectors are denoted by small bold letters and matrices (or tensors) are denoted by capital bold letters. For instance, $\mathbf{W}$ and $\mathbf{w}$ represent the weight tensor and its flattened version. Bar accent denotes the expectation over data points, e.g. $\bar{\mathbf{a}}$. Bracketed superscript $\mathbf{w}^{(\ell)}$ indicates the layer index. For a convolutional or a fully-connected layer, we mark its input and output vectors by $\mathbf{x}$ and $\mathbf{z}$. Thus given a feedforward neural network with $n$ layers, we can denote the forward process by

$$\mathbf{x}^{(\ell+1)} = h(\mathbf{z}^{(\ell)}) = h(\mathbf{W}^{(\ell)}\mathbf{x}^{(\ell)} + \mathbf{b}^{(\ell)}), \quad 1 \le \ell \le n, \tag{1}$$

where $h(\cdot)$ indicates the activation function (ReLU in this paper). For simplicity, we omit the analysis of bias $\mathbf{b}^{(\ell)}$ as it can be merged into activation. $||\cdot||_F$ denotes the Frobenius norm.

**Quantization Background** Uniform symmetric quantization maps the floating-point numbers to several fixed-points. These points (or grids) have the same interval and are symmetrically distributed. We denote the set that contains these grids as $\mathcal{Q}_b^{\text{u,sym}} = s \times \{-2^{b-1}, \ldots, 0, \ldots, 2^{b-1} - 1\}$. Here, $s$ is the step size between two grids and $b$ is the bit-width. Quantization function, denoted by $q(\cdot) : \mathcal{R} \to \mathcal{Q}_b^{\text{u,sym}}$, is generally designed to minimize the quantization error:

$$\min ||\hat{\mathbf{w}} - \mathbf{w}||_F^2. \text{ s.t. } \hat{\mathbf{w}} \in \mathcal{Q}_b^{\text{u,sym}} \tag{2}$$

Solving this minimization problem, one can easily get the $q(\cdot)$ by leveraging the rounding-to-nearest operation $\lfloor \cdot \rceil$. Rounding-to-nearest is a prevalent method to perform quantization, e.g. PACT (Choi et al., 2018). However, recently some empirical or theoretical evidence supports that simply minimizing the quantization error in parameter space does not bring optimal task performances. Specifically, Esser et al. (2020) propose to learn the step size $s$ by gradient descent in quantization-aware training (QAT). LAPQ (Nahshan et al., 2019) finds the optimal step size when the loss function is minimized without re-training the weights. Their motivations are all towards minimizing a final objective, which is the task loss, i.e.,

$$\min \mathbb{E}[L(\hat{\mathbf{w}})], \text{ s.t. } \hat{\mathbf{w}} \in \mathcal{Q}_b^{\text{u,sym}}. \tag{3}$$

While this optimization objective is simple and can be well-optimized in QAT scenarios, it is not easy to learn the quantized weight without end-to-end finetuning as well as sufficient training data and computing resources. In post-training quantization settings, we only have full precision weights that $\mathbf{w}^\star = \arg\min_{\mathbf{w}} \mathbb{E}[L(\mathbf{w})]$ where $\mathbf{w} \in \mathcal{R}$ and a small subset of training data to do calibration.

**Taylor Expansion** It turns out that the quantization imposed on weights could be viewed as a special case of weight perturbation. To quantitatively analyze the loss degradation caused by quantization, Nagel et al. (2020) use Taylor series expansions and approximates the loss degradation by

$$\mathbb{E}[L(\mathbf{w} + \Delta\mathbf{w})] - \mathbb{E}[L(\mathbf{w})] \approx \Delta\mathbf{w}^\mathsf{T}\bar{\mathbf{g}}^{(\mathbf{w})} + \frac{1}{2}\Delta\mathbf{w}^\mathsf{T}\bar{\mathbf{H}}^{(\mathbf{w})}\Delta\mathbf{w}, \tag{4}$$

where $\bar{\mathbf{g}}^{(\mathbf{w})} = \mathbb{E}[\nabla_{\mathbf{w}}L]$ and $\bar{\mathbf{H}}^{(\mathbf{w})} = \mathbb{E}[\nabla_{\mathbf{w}}^2 L]$ are the gradients and the Hessian matrix and $\Delta\mathbf{w}$ is the weight perturbation. Given the pre-trained model is converged to a minimum, the gradients can be safely thought to be close to $\mathbf{0}$. However, optimizing with the large-scale full Hessian is memory-infeasible on many devices as the full Hessian requires terabytes of memory space. To tackle this problem, they make two assumptions:

1. Layers are mutual-independent, thus the Hessian is in the form of layer-diagonal[1] and Kronecker-factored, i.e., $\bar{\mathbf{H}}^{(\mathbf{w}^{(\ell)})} = \mathbb{E}[\mathbf{x}^{(\ell)}\mathbf{x}^{(\ell),\mathsf{T}} \otimes \mathbf{H}^{(\mathbf{z}^{(\ell)})}]$, where $\otimes$ is the Kronecker product.
2. The second-order derivatives of pre-activations are constant diagonal matrix ($\mathbf{H}^{(\mathbf{z}^{(\ell)})} = c \times \mathbf{I}$) which is independent of input data points.

At last, the objective is transformed into a practical proxy signal, *the change in feature-maps ($\mathbf{z} = \mathbf{W}\mathbf{x}$)*, and the quantized model can be obtained by a layer-by-layer feature map reconstruction algorithm (with few calibration images). Recent works, like Bit-Split (Wang et al., 2020) and AdaQuant (Hubara et al., 2020), also take this layer-wise objective to improve the post-training quantization. However, they failed to quantize weights to INT2. We think the inherent reason is that when $\Delta\mathbf{w}$ grows higher, the former assumptions do not hold and an accurate signal is required.

## 3 PROPOSED METHOD

### 3.1 CROSS-LAYER DEPENDENCY

Denote the neural network output $\mathbf{z}^{(n)} = f(\theta)$, the loss function can be represented by $L(f(\theta))$ where $\theta = \mathrm{vec}[\mathbf{w}^{(1),\mathsf{T}}, \ldots, \mathbf{w}^{(n),\mathsf{T}}]^{\mathsf{T}}$ is the stacked vector of weights in all $n$ layers. The Hessian matrix can be computed by

$$\frac{\partial^2 L}{\partial \theta_i \partial \theta_j} = \frac{\partial}{\partial \theta_j}\left(\sum_{k=1}^{m}\frac{\partial L}{\partial \mathbf{z}_k^{(n)}}\frac{\partial \mathbf{z}_k^{(n)}}{\partial \theta_i}\right) = \sum_{k=1}^{m}\frac{\partial L}{\partial \mathbf{z}_k^{(n)}}\frac{\partial^2 \mathbf{z}_k^{(n)}}{\partial \theta_i \partial \theta_j} + \sum_{k,l=1}^{m}\frac{\partial \mathbf{z}_k^{(n)}}{\partial \theta_i}\frac{\partial^2 L}{\partial \mathbf{z}_k^{(n)}\partial \mathbf{z}_l^{(n)}}\frac{\partial \mathbf{z}_l^{(n)}}{\partial \theta_j}, \quad (5)$$

where $\mathbf{z}^{(n)} \in \mathcal{R}^m$. Since the pretrained full precision model is converged to a local minimum, we can assume the Hessian is positive-semidefinite (PSD). Specifically, the converged model has $\nabla_{\mathbf{z}^{(n)}} L$ close to $\mathbf{0}$ so the first term in Eq. (5) is neglected and Hessian becomes the Gauss-Newton (GN) matrix $\mathbf{G}^{(\theta)}$. GN matrix can be written in matrix form (Botev et al., 2017) as

$$\mathbf{H}^{(\theta)} \approx \mathbf{G}^{(\theta)} = \mathbf{J}_{\mathbf{z}^{(n)}}(\theta)^{\mathsf{T}}\mathbf{H}^{(\mathbf{z}^{(n)})}\mathbf{J}_{\mathbf{z}^{(n)}}(\theta), \quad (6)$$

where $\mathbf{J}_{\mathbf{z}^{(n)}}(\theta)$ is the Jacobian matrix of the network output with respect to the network parameters. However, in practice, we cannot explicitly compute and store the Jacobian for each input data point in such a raw form. To reduce the computation and memory budget, we will transform the second-order error into the network output, as shown in the following theorem.

**Theorem 3.1.** *Consider an $n$-layer feedforward neural network with ReLU activation function. Assuming all weights are quantized, the second-order error optimization can be transformed by:*

$$\underset{\hat{\theta}}{\arg\min}\, \Delta\theta^{\mathsf{T}}\bar{\mathbf{H}}^{(\theta)}\Delta\theta \approx \underset{\hat{\theta}}{\arg\min}\, \mathbb{E}\left[\Delta\mathbf{z}^{(n),\mathsf{T}}\mathbf{H}^{(\mathbf{z}^{(n)})}\Delta\mathbf{z}^{(n)}\right]. \quad (7)$$

**Remark 3.1.** The same transformation is also applicable for activation quantization. The quadratic loss is defined as $\mathbb{E}[\Delta\gamma^{\mathsf{T}}\mathbf{H}^{(\gamma)}\Delta\gamma]$ where $\Delta\gamma = \mathrm{vec}[\Delta\mathbf{x}^{(1),\mathsf{T}}, \ldots, \Delta\mathbf{x}^{(n),\mathsf{T}}]^{\mathsf{T}}$.

We prove the theorem using the quadratic form, details can be found in Appendix A.1. Here we provide a sketch of the proof by matrix form. The product of perturbation and Jacobian can be thought as the first-order Taylor approximation of the change in network output $\Delta\mathbf{z}^{(n)}$:

$$\Delta\mathbf{z}^{(n)} = \hat{\mathbf{z}}^{(n)} - \mathbf{z}^{(n)} \approx \mathbf{J}_{\mathbf{z}^{(n)}}(\theta)\Delta\theta. \quad (8)$$

Therefore, combining Eq. (8) and Eq. (6) we can transform the large-scale second-order error into the change in network outputs characterized by the output Hessian $\mathbf{H}^{(\mathbf{z}^{(n)})}$. The theorem indicates a simple observation, suppose a well-trained teacher model and an initialized student model, we can minimize their discrepancy by reconstructing the network's final output $\mathbf{z}^{(n)}$, which coincides with and generalizes the distillation (Hinton et al., 2015; Polino et al., 2018). LAPQ (Nahshan et al., 2019) also considers the dependency but their optimization does not rely on second-order information. However, we should emphasize that distillation requires the same computation and data resources as in normal training procedure, which is impractical for PTQ with limited data.

---

[1]To prevent ambiguity, we hereby use layer-diagonal Hessian to replace the common name "block-diagonal Hessian" because the block in this paper means a building block in the CNNs.

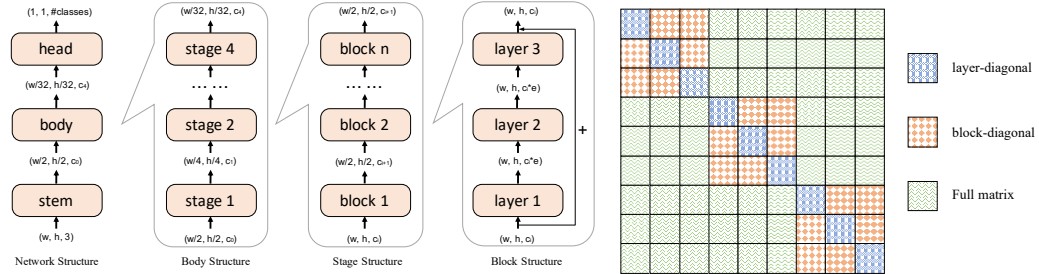

(a) A typical structure of CNN (taken from Radosavovic et al. (2020)). Network is composed of a stem layer (first convolution on input images), a body and a head layer (average pooling with a fully connected layer). A body contains several stages, and a stage contains several blocks. A representative block is the bottleneck block with residual path.

(b) An example illustration of Hessian (or Fisher) matrix. Blue sub-block means the layer-diagonal and each layer are mutual-independent; orange sub-block consider the dependency inside a building block and green parts measure all dependencies.

Figure 1: We define 4 kinds of reconstruction granularity, namely net-wise, stage-wise, block-wise and layer-wise optimization, each of them corresponds an essential component of CNN.

## 3.2 Block Reconstruction

Although the network output reconstruction has an accurate estimation of the second-order error, we find in practice it is worse than the layer-by-layer reconstruction in PTQ. The primary reason for this is optimizing the whole networks over 1024 calibration data samples leads to over-fitting easily. As Jakubovitz et al. (2019) explained, the networks can have perfect expressivity when the number of parameters exceeds the number of data samples during training, but lower training error does not ensure lower test error. We find layer-wise reconstruction acts like a regularizer which reduces the generalization error by matching each layer's output distribution. In other words, both layer-wise and network-wise output reconstruction has their own drawbacks. And there should be a better bias-variance trade-off choice to conduct reconstruction at an intermediate granularity.

The layer-wise optimization corresponds to layer-diagonal Hessian (Fig. 1b blue parts) and the network-wise optimization corresponds to full Hessian (Fig. 1b green parts). Similarly, we can define an intermediate block-diagonal Hessian. Formally, if layer $k$ to layer $\ell$ (where $1 \leq k < \ell \leq n$) form a block, the weight vector is defined as $\tilde{\theta} = \text{vec}[\mathbf{w}^{(k),\mathsf{T}}, \ldots, \mathbf{w}^{(\ell),\mathsf{T}}]^{\mathsf{T}}$ and the Hessian can be also transformed by $\Delta\tilde{\theta}^{\mathsf{T}}\bar{\mathbf{H}}^{(\tilde{\theta})}\Delta\tilde{\theta} = \mathbb{E}[\Delta\mathbf{z}^{(\ell),\mathsf{T}}\mathbf{H}^{(\mathbf{z}^{(\ell)})}\Delta\mathbf{z}^{(\ell)}]$. Such block-diagonal Hessian ignores the inter-block dependency and considers the intra-block dependency but it produces less generalization error. Then we can block-by-block reconstruct the intermediate output.

To this end, we define 2 extra kinds of intermediate *reconstruction granularity*: Stage-wise reconstruction and Block-wise reconstruction. These 4 reconstruction granularities are described below:

1. **Layer-wise Reconstruction:** Assume the Hessian matrix is layer-diagonal and optimize the layer output one-by-one. It does not consider cross-layer dependency and resemble existing methods (Nagel et al., 2020; Hubara et al., 2020; Wang et al., 2020).
2. **Block-wise Reconstruction:** A block is the core component in modern CNN, such as the Residual Bottleneck Block as shown in Fig. 1a. This method assumes the Hessian matrix is block-diagonal and block-by-block perform reconstruction, which ignores inter-block dependencies.
3. **Stage-wise Reconstruction:** A stage is where the featuremaps will be downsampled and generate more channels, which is believed to produce higher-level features. Typical CNN in ImageNet dataset contains 4 or 5 different stages. This method simultaneously optimizes all layers within a stage and thus considers more dependencies than the block-wise method.
4. **Network-wise Reconstruction:** Optimize the whole quantized network by reconstructing the output of the final layers. This method resembles distillation but does not result in good performances with few images because of high generalization error.

The relationship between network, stage, block, and layer is illustrated in Fig. 1a. *We test these 4 kinds of reconstruction granularity and find that block-wise optimization outperforms others.* We think this is because the main off-diagonal loss in the Hessian is concentrated in each block, as Fig. 1b orange part illustrated, while the inter-block loss is small and can be ignored in the opti-

---

**Algorithm 1:** BRECQ optimization

---

**Input:** Pretrained FP model; Calibration dataset, iteration $T$

**for** *all $i = 1, 2, \ldots, N$-th block in the FP model* **do**

    Collect input data to the block $\mathbf{x}^{(i)}$, the FP output $\mathbf{z}^{(i)}$ and its gradient $\mathbf{g}^{(\mathbf{z}^{(i)})}$ ;

    **for** *all $j = 1, 2, \ldots, T$-iteration* **do**

        Get quantized output $\hat{\mathbf{z}}^{(i)}$ and compute $\Delta\mathbf{z}^{(i)} = \mathbf{z}^{(i)} - \hat{\mathbf{z}}^{(i)}$;

        Descend Eq. (10) and update the rounding of all the weights in this block (Eq. (16));

        **if** *Activation Quantization is triggered* **then**

            Update the activation quantization step size (Eq. (18)).

    After optimization, compute the sensitivity for each layer and between layers (2-bit only);

**return** Quantized model, Sensitivities for mixed precision;

---

mization. The shortcut connections, which is proposed in (He et al., 2016), may also increase the dependencies within a block. Also, the stage-wise or net-wise optimization suffer from the bad generalization on the validation set and degenerate the final performances. We report the quantitative comparison in Sec. 4.1. We name our algorithm BRECQ, because we choose block as our base reconstruction unit. It is necessary to point out that our analysis does not give the optimal configuration of the reconstruction granularity. The choice of block-wise optimization comes from our experiments and we find this choice has two merits. (1) No hyper-parameters included and (2) applicable for all models and all tasks we tested.

## 3.3 APPROXIMATING PRE-ACTIVATION HESSIAN

With block-diagonal approximated Hessian matrix, we can measure the cross-layer dependency inside each block and transform any block's second-order error to the output of this block $\mathbb{E}[\Delta\mathbf{z}^{(\ell),\mathsf{T}}\mathbf{H}^{(\mathbf{z}^{(\ell)})}\Delta\mathbf{z}^{(\ell)}]$. This objective requires the further computation of the knowledge in the rest of the network, i.e., pre-activation Hessian $\mathbf{H}^{(\mathbf{z}^{(\ell)})}$. One way is to follow Nagel et al. (2020) and assume $\mathbf{H}^{(\mathbf{z}^{(\ell)})} = c \times \mathbf{I}$. Therefore the quadratic loss becomes $||\Delta\mathbf{z}^{(\ell)}||^2$. This method might be easy to implement but lose too much information.

We use the diagonal Fisher Information Matrix (FIM) to replace the pre-activation Hessian. Formally, given a probabilistic model $p(x|\theta)$, the FIM is defined as:

$$\bar{\mathbf{F}}^{(\theta)} = \mathbb{E}\left[\nabla_\theta \log p_\theta(y|x)\nabla_\theta \log p_\theta(y|x)^\mathsf{T}\right] = -\mathbb{E}\left[\nabla^2_\theta \log p_\theta(y|x)\right] = -\bar{\mathbf{H}}^{(\theta)}_{\log p(x|\theta)}. \quad (9)$$

The FIM is equal to the negative expected Hessian of the log-likelihood function, therefore, a simple corollary is that the Hessian of task loss will become FIM if the model distribution matches the true data distribution (LeCun et al., 2012). Although matching true data distribution seems impossible, this is the best we can do since the pretrained model is converged.

The diagonal of the pre-activation FIM is equal to the squared gradients of each elements, which is successfully used in Adam (Kingma & Ba, 2014) for the second momentum. The optimization objective becomes

$$\min_{\hat{\mathbf{w}}} \mathbb{E}\left[\Delta\mathbf{z}^{(\ell),\mathsf{T}}\mathbf{H}^{(\mathbf{z}^{(\ell)})}\Delta\mathbf{z}^{(\ell)}\right] = \min_{\hat{\mathbf{w}}} \mathbb{E}\left[\Delta\mathbf{z}^{(\ell),\mathsf{T}}\mathrm{diag}\left((\frac{\partial L}{\partial \mathbf{z}^{(\ell)}_1})^2, \ldots, (\frac{\partial L}{\partial \mathbf{z}^{(\ell)}_a})^2\right)\Delta\mathbf{z}^{(\ell)}\right]. \quad (10)$$

Compared with the MSE minimization, the above minimization incorporates the squared gradient information. If the output has higher absolute gradients, it will receive more attention when being reconstructed. A similar method for pruning the pre-activation has been proposed in Theis et al. (2018).

Note that BRECQ is compatible with any optimization method, like STE (Hubara et al., 2017). Here we adopt adaptive rounding (Nagel et al., 2020) for weights and learned step size (Esser et al., 2020) for activation step size because we observe they generally perform better in PTQ, see details in Appendix B.4.1. We formulate the overall calibration algorithm for a unified precision model in algorithm 1. We should emphasize that we only need a small subset (1024 in our experiments) of the

whole training dataset to calibrate the quantized model. And we can obtain a quantized ResNet-18 within 20 minutes on a single GTX 1080TI GPU.

## 3.4 MIXED PRECISION

To further push the limit of post-training quantization, we employ mixed precision techniques, which can be formulated by

$$\min_{\mathbf{c}} L(\hat{\mathbf{w}}, \mathbf{c}), \ \ \text{s.t.} \ H(\mathbf{c}) \leq \delta, \ \ \mathbf{c} \in \{2, 4, 8\}^n. \tag{11}$$

Here, $\mathbf{c}$ is the bit-width vector with the shape of number of layers. $H(\cdot)$ is a hardware performance measurement function, which is used to ensure the mixed precision model has the same or lower hardware performance (e.g., memory and speed) than a predefined threshold $\delta$. We choose 2, 4, 8-bit for mixed precision because they are most common in practical deployment.

Regarding the training loss $L$, we find that nearly all existing literature (Cai et al., 2020; Hubara et al., 2020; Dong et al., 2019) uses layer-wise measurement. They all assume the sensitivity within a layer is independent and can be summed together. Therefore, the mixed precision problem becomes an integer programming problem. However, we argue that the loss measurement should contain two parts: diagonal loss and off-diagonal loss, the first is the same with previous works and measure the sensitivity of each layer independently, while the off-diagonal loss is used to measure the cross-layer sensitivity. Theoretically, we should examine all permutations, which results in $3^n$ possibilities and prohibits the search algorithm. Our first attempt is to reduce the off-diagonal loss into the block-level as we mentioned that the Hessian can be approximated to a block-diagonal matrix. Granted, we still find the search space is large, for example, if a block has four layers, then we have to consider the $3^4 = 81$ permutations for a single block. Based on our preliminary experiments, we find that 4-bit and 8-bit quantization nearly do not drop the final accuracy. Hence we only take 2-bit permutations into consideration and drastically reduce the search space. We use genetic algorithm (Guo et al., 2020) to search the optimal bitwidth configuration with hardware performance threshold, the algorithm is located in algorithm 2. Due to space limits, we put related works in Appendix 5. Readers can refer to related works for a brief discussion on quantization and second-order analysis.

## 4 EXPERIMENTS

In this section, we report experimental results for the ImageNet classification task and MS COCO object detection task. The detailed implementation of the experiments can be found in the Appendix B.4.4. The rest of this section will contain ablation study on reconstruction granularity, classification and detection results, mixed precision results and comparison with quantization-aware training. In Appendix B, we conduct more experiments, including the impact of the first and the last layer, the impact of calibration dataset size and data source.

## 4.1 ABLATION STUDY

We test four kinds of reconstruction granularity: Net-wise, Stage-wise, Block-wise, and Layer-wise reconstruction. We conduct ImageNet experiments using MobileNetV2 and ResNet-18 with 2-bit weight quantization for all layers except for the first and the last layer. It can be seen from Table 1 that Block-wise optimization outperforms other methods. This result implies that the generalization error in net-wise and stage-wise optimization outweighs their off-diagonal loss. In ResNet-18, we find the difference is not significant, this can be potentially attributed to that ResNet-18 only has 19 layers in the body and the block size, as well as the stage size, is small, therefore leading to indistinct results.

Table 1: Ablation study.

| Model | Layer | Block | Stage | Net |
|---|---|---|---|---|
| ResNet-18 | 65.19 | **66.39** | 66.01 | 54.15 |
| MobileNetV2 | 52.13 | **59.67** | 54.23 | 40.76 |

## 4.2 IMAGENET

We conduct experiments on a variety of modern deep learning architectures, including ResNet (He et al., 2016) with normal convolution, MobileNetV2 (Sandler et al., 2018) with depthwise separa-

Table 2: Accuracy comparison on *weight-only quantized* post-training models. Activations here are unquantized and kept full precision. We also conduct variance study for our experiments. Bold values indicates best results. * indicates our implementation based on open-source codes.

| Methods | Bits (W/A) | ResNet-18 | ResNet-50 | MobileNetV2 | RegNet-600MF | RegNet-3.2GF | MnasNet-2.0 |
|---|---|---|---|---|---|---|---|
| Full Prec. | 32/32 | 71.08 | 77.00 | 72.49 | 73.71 | 78.36 | 76.68 |
| Bias Correction* | 4/32 | 50.43 | 64.64 | 62.82 | 67.09 | 71.73 | 72.31 |
| OMSE (Choukroun et al., 2019) | 4/32 | 67.12 | 74.67 | - | - | - | - |
| AdaRound (Nagel et al., 2020) | 4/32 | 68.71 | 75.23 | 69.78 | 71.97* | 77.12* | 74.87* |
| AdaQuant (Hubara et al., 2020) | 4/32 | 68.82 | 75.22 | 44.78 | - | - | - |
| Bit-Split (Wang et al., 2020) | 4/32 | 69.11 | 75.58 | - | - | - | - |
| BRECQ (Ours) | 4/32 | $70.70_{\pm 0.07}$ | $76.29_{\pm 0.04}$ | $71.66_{\pm 0.04}$ | $73.02_{\pm 0.09}$ | $78.04_{\pm 0.04}$ | $76.00_{\pm 0.02}$ |
| Bias Correction* | 3/32 | 12.85 | 7.97 | 10.89 | 28.82 | 17.95 | 40.72 |
| AdaRound (Nagel et al., 2020)* | 3/32 | 68.07 | 73.42 | 64.33 | 67.71 | 72.31 | 69.33 |
| AdaQuant (Hubara et al., 2020)* | 3/32 | 58.12 | 67.61 | 12.56 | - | - | - |
| Bit-Split (Wang et al., 2020) | 3/32 | 66.75 | 73.24 | - | - | - | - |
| BRECQ (Ours) | 3/32 | $69.81_{\pm 0.05}$ | $75.61_{\pm 0.09}$ | $69.50_{\pm 0.12}$ | $71.48_{\pm 0.07}$ | $77.22_{\pm 0.04}$ | $74.58_{\pm 0.08}$ |
| Bias Correction* | 2/32 | 0.13 | 0.12 | 0.14 | 0.18 | 0.11 | 0.11 |
| AdaRound (Nagel et al., 2020)* | 2/32 | 55.96 | 47.95 | 32.54 | 25.66 | 24.70 | 30.60 |
| AdaQuant (Hubara et al., 2020)* | 2/32 | 0.30 | 0.49 | 0.11 | - | - | - |
| BRECQ (Ours) | 2/32 | $66.30_{\pm 0.12}$ | $72.40_{\pm 0.12}$ | $59.67_{\pm 0.13}$ | $65.83_{\pm 0.13}$ | $73.88_{\pm 0.14}$ | $67.13_{\pm 0.13}$ |

Table 3: Accuracy comparison on *fully quantized* post-training models. Activations here are quantized to 4-bit. Notations follows the upper table.

| Methods | Bits (W/A) | ResNet-18 | ResNet-50 | MobileNetV2 | RegNet-600MF | RegNet-3.2GF | MNasNet-2.0 |
|---|---|---|---|---|---|---|---|
| Full Prec. | 32/32 | 71.08 | 77.00 | 72.49 | 73.71 | 78.36 | 76.68 |
| ACIQ-Mix (Banner et al., 2019) | 4/4 | 67.0 | 73.8 | - | - | - | - |
| ZeroQ (Cai et al., 2020)* | 4/4 | 21.71 | 2.94 | 26.24 | 28.54 | 12.24 | 3.89 |
| LAPQ (Nahshan et al., 2019) | 4/4 | 60.3 | 70.0 | 49.7 | 57.71* | 55.89* | 65.32* |
| AdaQuant (Hubara et al., 2020) | 4/4 | 67.5 | 73.7 | 34.95* | - | - | - |
| Bit-Split (Wang et al., 2020) | 4/4 | 67.56 | 73.71 | - | - | - | - |
| BRECQ (Ours) | 4/4 | $69.60_{\pm 0.04}$ | $75.05_{\pm 0.09}$ | $66.57_{\pm 0.67}$ | $68.33_{\pm 0.28}$ | $74.21_{\pm 0.19}$ | $73.56_{\pm 0.24}$ |
| ZeroQ (Cai et al., 2020)* | 2/4 | 0.08 | 0.08 | 0.10 | 0.10 | 0.05 | 0.12 |
| LAPQ (Nahshan et al., 2019)* | 2/4 | 0.18 | 0.14 | 0.13 | 0.17 | 0.12 | 0.18 |
| AdaQuant (Hubara et al., 2020)* | 2/4 | 0.21 | 0.12 | 0.10 | - | - | - |
| BRECQ (Ours) | 2/4 | $64.80_{\pm 0.08}$ | $70.29_{\pm 0.23}$ | $53.34_{\pm 0.15}$ | $59.31_{\pm 0.49}$ | $67.15_{\pm 0.11}$ | $63.01_{\pm 0.35}$ |

ble convolution and RegNet (Radosavovic et al., 2020) with group convolution. Last but not least important, we also investigate the neural architecture searched (NAS) models, MNasNet (Tan et al., 2019). In Table 2, we only quantize weights into low-bit integers and keep activations full precision. We compare with strong baselines including Bias Correction, optimal MSE, AdaRound, AdaQuant, and Bit-split. Note that the first and the last layer are kept with 8-bit. While most of the existing methods have good performances in 4-bit quantization, they cannot successfully quantize the model into 2-bit. Our method consistently achieves the lowest accuracy degradation for ResNets (within 5%) and other compact models. We further quantize activations into 4-bit to make the quantized model run on integer-arithmetic hardware platforms. We find that 4-bit activation quantization can have a huge impact on RegNet and MobileNet. Nonetheless, our methods produce higher performance than other state-of-the-arts. To be noted, BRECQ is the first to promote the 2W4A accuracy of PTQ to a usable level while all other existing methods crashed.

## 4.3 COMPARISON WITH QUANTIZATION-AWARE TRAINING

Table 4: Performance as well as training cost comparison with quantization-aware training (QAT).

| Models | Methods | Precision | Accuracy | Model Size | Training Data | GPU hours |
|---|---|---|---|---|---|---|
| ResNet-18 FP: 71.08 | ZEROQ (CAI ET AL., 2020)) | 4/4 | 21.20 | 5.81 MB | **0** | **0.008** |
| | BRECQ (OURS) | 4/4 | **69.60** | 5.81 MB | 1024 | 0.4 |
| | BRECQ (W/ DISTILLED DATA) | 4/4 | **69.32** | 5.81 MB | 0 | 0.4 |
| | PACT (CHOI ET AL., 2018) | 4/4 | 69.2 | 5.81 MB | 1.2 M | 100 |
| | DSQ (GONG ET AL., 2019) | 4/4 | 69.56 | 5.81 MB | 1.2 M | 100 |
| | LSQ (ESSER ET AL., 2020) | 4/4 | **71.1** | 5.81 MB | 1.2 M | 100 |
| MobileNetV2 FP: 72.49 | BRECQ (OURS) | 4/4 | **66.57** | 2.26 MB | 1024 | 0.8 |
| | PACT (CHOI ET AL., 2018) | 4/4 | 61.40 | 2.26 MB | 1.2 M | 192 |
| | DSQ (GONG ET AL., 2019) | 4/4 | 64.80 | 2.26 MB | 1.2 M | 192 |
| | BRECQ (OURS) | Mixed/8 | **70.74** | 1.38 MB | **1024** | **3.2** |
| | HAQ (WANG ET AL., 2019) | Mixed/8 | **70.90** | 1.38 MB | 1.2 M | 384 |

Table 5: Objection detection task (MS COCO) comparison on *fully quantized* post-training models. Activations here are quantized to 8-bit. We report the bounding box mean Average Precision (mAP) metric.

| Models | Backbone | Full Prec. | Bias Correction* | | AdaRound* | | ZeroQ | BRECQ (Ours) | | |
|---|---|---|---|---|---|---|---|---|---|---|
| | | 32/32 | 8/8 | 4/8 | 4/8 | 2/8 | 4$_{MP}$/8 | 8/8 | 4/8 | 2/8 |
| Faster RCNN | ResNet-18 | 34.55 | 34.30 | 0.84 | 33.96 | 23.01 | - | **34.53** | **34.34** | **31.82** |
| (Ren et al., | ResNet-50 | 38.55 | 38.25 | 0.25 | 37.58 | 19.63 | - | **38.54** | **38.29** | **34.23** |
| 2015) | MobileNetV2 | 33.44 | 33.24 | 18.39 | 32.77 | 16.35 | - | **33.40** | **33.18** | **27.54** |
| RetinaNet | ResNet-18 | 33.20 | 33.00 | 0.04 | 32.59 | 19.93 | - | **33.14** | **33.01** | **31.42** |
| (Lin et al., | ResNet-50 | 36.82 | 36.68 | 0.07 | 36.00 | 19.97 | 33.7 | **36.73** | **36.65** | **34.75** |
| 2017) | MobileNetV2 | 32.63 | **32.60** | 18.47 | 31.89 | 14.10 | - | 32.57 | **32.31** | **27.59** |

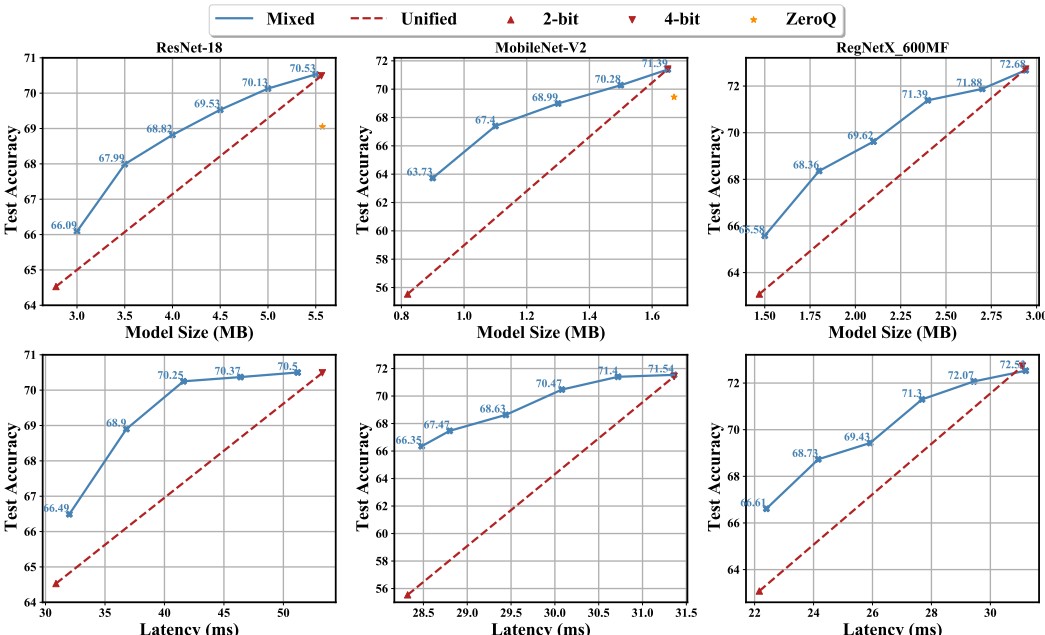

Figure 2: Mixed Precision results.

In this section, we compare our algorithm (post-training quantization) with some quantization-aware training methods, including PACT (Choi et al., 2018), DSQ (Gong et al., 2019), LSQ (Esser et al., 2020), and a mixed precision technique HAQ (Wang et al., 2019). Table 4 shows that although BRECQ is a PTQ method with limited available data, it can achieve comparable accuracy results with existing quantization-aware training models. In addition, our method can surpass them in 4-bit MobileNetV2 while using less than one training GPU hours. Our method also has comparable accuracy with HAQ, which is a training-based mixed precision method. Note that our GPU hours include 3 unified precision training (2-, 4-, 8-bit respectively) and further mixed-precision training only needs to check the lookup table. Instead, HAQ would end-to-end search for each hardware performance threshold from scratch.

## 4.4   MS COCO

To validate the effectiveness of BRECQ on other tasks, we conduct object detection on the two-stage Faster-RCNN (Ren et al., 2015) and the one-stage RetinaNet (Lin et al., 2017). ResNet-18, 50 as well as MobileNetV2 are adopted as backbones for the detection model. Results in Table 5 demonstrate our method nearly does not drop the performance in 4-bit weight quantization and 8-bit activation. In particular, BRECQ only decreases **0.21%** mAP performance on 4-bit ResNet-18 backboned Faster RCNN. On 4-bit ResNet-50 backboned RetinaNet, our method is outperforms the mixed precision based ZeroQ model by **3%** mAP. Even when the weight bit decreases to 2, the model still achieves near-to-original mAP.

## 4.5 MIXED PRECISION

In this section, we test *(1) model-size guaranteed mixed precision* and *(2) FPGA latency guaranteed mixed precision*[2] to unleash the potential of mixed precision and further push the limit of PTQ. We choose ResNet-18, MobileNetV2, and RegNetX-600MF to validate the efficacy of our algorithm. Note that in this section, we keep activation in 8-bit because we only compare the discrepancy between the unified and mixed precision in weights. We omit 3-bit weight quantization in unified precision because it is usually unfriendly to the hardware. Latency settings can be found in Appendix B.4.3. From Fig. 2 we find that (1) mixed precision consistently outperforms unified precision, especially when using extremely low-bit, e.g., up to 10% accuracy increase with the same latency as the 2-bit model. (2) mixed precision can produce many bit configurations that can adapt to plenty of hardware requirements while unified precision can only have 2 fixed models.

## 5 RELATED WORKS

**Quantization** Model quantization can be classified into two categories: Quantization-aware Training (QAT) and Post-training Quantization (PTQ). Rounding floating-point numbers to fixed-points numbers will produce 0 gradients almost everywhere. Therefore, most QAT methods employ the Straight-Through Estimator (STE) for gradients approximation. Gong et al. (2019) uses a differentiable tanh function to gradually approach the step function. Choi et al. (2018); Esser et al. (2020) introduces parameterized clipping thresholds to learn it by STE. Apart from uniform quantization, some works like Li et al. (2019) argue that non-uniform quantization has better performance than uniform quantization while keeping its efficiency. Despite the promising results given by QAT methods, they usually need more than 100 GPU hours to get it. In that case, PTQ plays an important role which is what we focus on in this paper. Generally, most deep learning models can be safely quantized to 8-bit without re-training. Data-Free Quantization Nagel et al. (2019) even do layer-wise 8-bit PTQ without any data. However, in 4-bit quantization, most parameter space-based methods cannot obtain good performances. Recently, Nagel et al. (2020) propose to do layer-wise calibration and made huge progress in 4-bit quantization. Our work continues its analysis on Taylor expansion and considers the off-diagonal loss. Another perspective of quantification is the precision allocation scheme. Hardware-aware Quantization (HAQ Wang et al. (2019)) leverages reinforcement learning to search the optimal bitwidth configuration. Hessian-aware Weight Quantization (HAWQ) (Dong et al., 2019) utilizes the second-order information to decide the bitwidth. Mixed precision also appears in PTQ, such as the Pareto frontier method in ZeroQ (Cai et al., 2020) and the Integer Programming method in AdaQuant (Hubara et al., 2020).

**Second-order Analysis and Optimization** The history of second-order information in perturbation analysis can be traced to the 1990s like Optimal Brain Surgeon (Hassibi & Stork, 1993; Dong et al., 2017). The Hessian matrix is essential for pruning and quantization. As aforementioned, HAWQ uses the largest eigenvalue of Hessian to determine the sensitivity. Hessian matrix is also important for second-order optimization like Newton's method as it consists of the curvature information. However, calculating the real full Hessian is prohibitive on today's deep learning architectures. Therefore, approximations are made to simplify the calculation and make the storage more flexible, e.g., Gauss-Newton optimization with Kronecker-factored recursive approximation Botev et al. (2017). Hessian-Free optimization (Martens, 2010) avoids the explicit computation of the Hessian matrix by solving the linear system $g = Hv$. Second-order optimization with FIM is called Natural Gradient Descent (Amari, 1998). K-FAC (Martens & Grosse, 2015) utilizes the layer-diagonal FIM and the approximation of the expected Kronecker product to compute the curvature information.

## 6 CONCLUSION

In this paper, we propose BRECQ, a post-training quantization framework by analyzing the second-order error. We show that the reconstruction of quantization at the block granularity arrives at a good balance of cross-layer dependency and first order approximation, especially in 2-bit weight quantization where no prior works succeed to quantize. BRECQ is compatible with mixed precision and can reduce the search cost. To our best knowledge, BRECQ reaches the highest performance in post-training quantization and is the first to be on a par with quantization-aware training using 4-bit.

---

[2]We also test mobile CPU latency guaranteed mixed precision, located in Appendix B.3.

## ACKNOWLEDGMENT

We thank Markus Nagel and anonymous reviewers for their kind help of this work. This project is primarily supported by NSFC 61876032.

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

## A MAIN PROOFS

### A.1 PROOF OF THEOREM 3.1

*Proof.* We will prove the theorem using quadratic form. Denote the weight vector shape as $\theta \in \mathbb{R}^d$, and the network output vector shape as $\mathbf{z}^{(n)} \in \mathbb{R}^m$. The quadratic form of the $\Delta\theta^\mathsf{T}\mathbf{H}^{(\theta)}\Delta\theta$ can be represented by:

$$\Delta\theta^\mathsf{T}\mathbf{H}^{(\theta)}\Delta\theta = \sum_{i=1}^d \Delta\theta_i^2 \left(\frac{\partial^2 L}{\partial\theta_i^2}\right) + 2\sum_{i<j}^d \Delta\theta_i\Delta\theta_j \frac{\partial L}{\partial\theta_i\theta_j} = \sum_{i=1}^d\sum_{j=1}^d \left(\Delta\theta_i\Delta\theta_j\frac{\partial L}{\partial\theta_i\theta_j}\right), \quad (12)$$

where $L$ is the cross-entropy loss. Based on Eq. (5), we have

$$\frac{\partial^2 L}{\partial\theta_i\theta_j} = \sum_{k,l}^m \frac{\partial\mathbf{z}_k^{(n)}}{\partial\theta_l}\frac{\partial^2 L}{\partial\mathbf{z}_k^{(n)}\mathbf{z}_l^{(n)}}\frac{\partial\mathbf{z}_l^{(n)}}{\partial\theta_j} \tag{13}$$

Substituting above equation in Eq. (12), we have

$$\Delta\theta^\mathsf{T}\mathbf{H}^{(\theta)}\Delta\theta = \sum_{i=1}^d\sum_{j=1}^d \Delta\theta_i\Delta\theta_j \left(\sum_{k=1}^m\sum_{l=1}^m \frac{\partial\mathbf{z}_k^{(n)}}{\partial\theta_i}\frac{\partial^2 L}{\partial\mathbf{z}_k^{(n)}\mathbf{z}_l^{(n)}}\frac{\partial\mathbf{z}_l^{(n)}}{\partial\theta_j}\right) \tag{14a}$$

$$= \sum_{i=1}^d\sum_{j=1}^d\sum_{k=1}^m\sum_{l=1}^m \left(\Delta\theta_i\Delta\theta_j\frac{\partial\mathbf{z}_k^{(n)}}{\partial\theta_i}\frac{\partial^2 L}{\partial\mathbf{z}_k^{(n)}\mathbf{z}_l^{(n)}}\frac{\partial\mathbf{z}_l^{(n)}}{\partial\theta_j}\right) \tag{14b}$$

$$= \sum_{k=1}^m\sum_{l=1}^m \left(\frac{\partial^2 L}{\partial\mathbf{z}_k^{(n)}\mathbf{z}_l^{(n)}}\right)\left(\sum_{i=1}^d \Delta\theta_i\frac{\partial\mathbf{z}_k^{(n)}}{\partial\theta_i}\right)\left(\sum_{j=1}^d \Delta\theta_j\frac{\partial\mathbf{z}_k^{(n)}}{\partial\theta_j}\right) \tag{14c}$$

$$= (\Delta\theta\mathbf{J}\left[\frac{\mathbf{z}^{(n)}}{\theta}\right])^\mathsf{T}\mathbf{H}^{(\mathbf{z}^{(n)})}(\Delta\theta\mathbf{J}\left[\frac{\mathbf{z}^{(n)}}{\theta}\right]), \tag{14d}$$

where we define the $\mathbf{J}\left[\frac{x}{y}\right]$ is the Jacobian matrix of $x$ w.r.t. $y$. To this end, we use the first-order Taylor expansion as we did in Eq. (8) to approximate the change in network output, i.e.,

$$\Delta\mathbf{z}^{(n)} \approx \Delta\theta\mathbf{J}\left[\frac{\mathbf{z}^{(n)}}{\theta}\right] \tag{15}$$

Therefore, the final objective is transformed to $\Delta\mathbf{z}^{(n),\mathsf{T}}\mathbf{H}^{(\mathbf{z}^{(n)})}\Delta\mathbf{z}^{(n)}$. □

# B EXPERIMENTS

## B.1 EFFECT OF THE FIRST AND THE LAST LAYER

Many papers claim that the first and the last layer can have a huge impact on the final accuracy. In this section, we investigate this phenomenon as well as the impact of the first and the last layer on hardware performances. We test ResNet-18, MobileNetV2 as well as RegNet-600MF. Our observations include:

1. In terms of accuracy, the 4-bit quantization is essentially good, both of these two layers won't drop too much accuracy (with 0.2%). But in 2-bit quantization, the last fully connected layer is far more important than the first layer. We also observe that the first layer in MobileNetV2 and RegNet (3×3 kernels, 32 channels) is slightly more sensitive than that in ResNet-18 (7×7 kernel, 64 channels).
2. In terms of model size, the first layer merely has a minor impact because the input images only have 3 channels, while the last layer contains many weight parameters and greatly affects the memory footprint. We should point out that just the model size in the first layer is low doesn't mean the memory burden is low, because the input image will cost huge memory space.
3. In terms of latency, the situation depends on the architecture. For example, in ResNet-18 the first layer has a huge impact on the latency, while in MobileNetV2 and RegNet-600MF the last layer is more important than the first layer. This is because the latency is affected by multiple factors, such as the input size of the featuremap, the FLOPs, and the weight memory size. The arithmetic intensity (OPs/byte) greatly affects the latency. We find that the operations with high arithmetic intensity, i.e., shallow layers in the network, generate a less latency gap between different bit-widths.

In conclusion, we find that keeping the first and the last layer 8-bit is unnecessary. Especially in ResNet-18, we find that setting all layers to 4-bit results in 53.3 ms latency and is faster than the 59.8 ms in 2-bit quantization (with first and last layer 8-bit), but the accuracy is even 4% higher. Such phenomenon indicates the potential power of the mixed precision.

## B.2 EFFECT OF DATA

We evaluated the influence of the size of calibration dataset and the source of the data on ResNet-18. We test different numbers of input data point and find that the improvement in 4-bit quantization is trivial. Yet in 2-bit quantization we can see that the accuracy increases 5% when #data points increase. We also test the distilled data introduced in ZeroQ (Cai et al., 2020). Distilled data is learned from pretrained models' BN statistics, i.e., $\mathbf{x}^1_{\text{distilled}} = \arg\min_{\mathbf{x} \in \mathcal{R}} \sum_{i=1}^{n} ((\mu_i - \hat{\mu}_i)^2 + (\varsigma_i - \hat{\varsigma}_i))$ where $\mu_i$ and $\varsigma_i$ is the original mean and variance in BN statistics of the $(i)$-th layer. We find that distilled data performs good in 4-bit quantization but still has a large margin with the original ImageNet dataset in 2-bit quantization. We also find the final accuracy does not benefit much from the increase of number of distilled data, this might because the distilled data are minimized by a same objective and has low diversity.

Table 6: Impact of the first and the last layer

| Models | No Quantization | | Precision 4/8 | | | Precision 2/8 | | |
|---|---|---|---|---|---|---|---|---|
| | First | Last | Accuracy | Model Size | Latency | Accuracy | Model Size | Latency |
| ResNet-18 FP: 71.08 | ✓ | ✓ | 70.76 | 5.81 MB | 70.72 ms | 66.30 | 3.15 MB | 59.84 ms |
| | ✗ | ✓ | 70.66 | 5.81 MB | 53.76 ms | 65.95 | 3.15 MB | 31.20 ms |
| | ✓ | ✗ | 70.64 | 5.57 MB | 70.08 ms | 64.87 | 2.79 MB | 58.72 ms |
| | ✗ | ✗ | 70.58 | 5.56 MB | 53.28 ms | 64.53 | 2.78 MB | 30.88 ms |
| MobileNetV2 FP: 72.49 | ✓ | ✓ | 71.80 | 2.26 MB | 32.80 ms | 59.59 | 1.74 MB | 30.40 ms |
| | ✗ | ✓ | 71.69 | 2.26 MB | 32.64 ms | 59.13 | 1.74 MB | 30.24 ms |
| | ✓ | ✗ | 71.42 | 1.65 MB | 31.52 ms | 56.29 | 0.83 MB | 28.48 ms |
| | ✗ | ✗ | 71.42 | 1.65 MB | 31.36 ms | 55.58 | 0.82 MB | 28.32 ms |
| RegNet-600MF FP: 73.71 | ✓ | ✓ | 72.98 | 3.19 MB | 31.84 ms | 65.66 | 1.84 MB | 23.20 ms |
| | ✗ | ✓ | 72.89 | 3.19 MB | 31.68 ms | 65.83 | 1.85 MB | 22.88 ms |
| | ✓ | ✗ | 72.69 | 2.94 MB | 31.20 ms | 62.93 | 1.47 MB | 22.40 ms |
| | ✗ | ✗ | 72.73 | 2.94 MB | 31.04 ms | 63.08 | 1.47 MB | 22.08 ms |

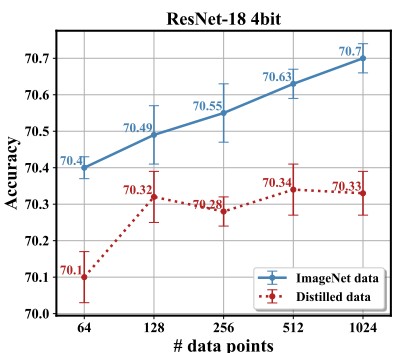
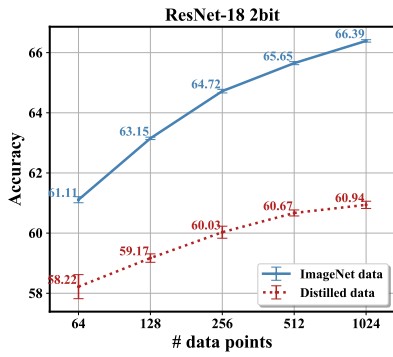

Figure 3: Effect of #data points and data source.

## B.3 Mobile CPU Latency Guaranteed Mixed Precision

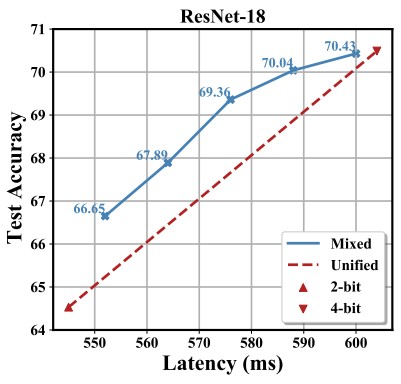
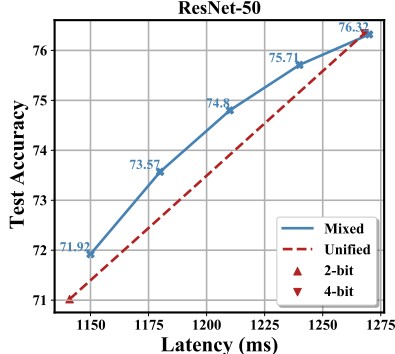

Figure 4: Mixed precision results on ResNet-18 and 50.

In this section, we test the mobile CPU latency guaranteed mixed precision. The latency lookup table is tested using the technique in Gong et al. (2019). We only validate it on ResNet-18 and ResNet-50 because the current low-bit General Matrix Multiply (GEMM) implementation only supports normal convolution. The results concur with Fig. 2. Below 4-bit, the mixed precision can achieve better task performance than the unified precision models. For ResNet-50, the improvement is lower than that for ResNet-18 and any other mixed precision models. We think this is because the sensitivity in ResNet-50 is not distinct and therefore the improvement brought by mixed precision is trivial.

## B.4 Implementation

### B.4.1 Learning Strategies

In this work, we mainly focus on developing optimization objective rather than optimization strategies. We observe adaptive rounding performs well in post-training quantization. A brief introduction on AdaRound is given below, the detailed algorithm can be found in Nagel et al. (2020).

Traditional quantization function is performed by rounding-to-nearest operation: $\hat{\mathbf{w}} = s \times \text{clip}(\lfloor \mathbf{w}/s \rceil, n, p)$. AdaRound optimizes the rounding policy in post-training quantization. Specifically, all weights are initially rounded by floor operation, and a learnable variable $\mathbf{v}$ determines the final rounding result to be flooring or ceiling. A sigmoid-like function $\sigma(\cdot)$ keeps the learnable variable $\mathbf{v}$ moving between 0 and 1 and a regularization term assures the $\sigma(\mathbf{v})$ can converged to either 0 or 1. The formulation is given by

$$\hat{\mathbf{w}} = s \times \text{clip}\left(\lfloor \frac{\mathbf{w}}{s} \rfloor + \sigma(\mathbf{v}), n, p\right) \tag{16}$$

The minimization problem together with the regularization is given by

$$\arg\min_{\mathbf{v}} \mathbb{E}\left[\Delta\mathbf{z}^{(\ell),\mathsf{T}}\text{diag}((\frac{\partial L}{\partial \mathbf{z}_1^{(\ell)}})^2, \dots, (\frac{\partial L}{\partial \mathbf{z}_a^{(\ell)}})^2)\Delta\mathbf{z}^{(\ell)}\right] + \lambda \sum_i \left(1 - |2\sigma(\mathbf{v}_i) - 1|^\beta\right), \tag{17}$$

where progressively decreasing $\beta$ in the calibration ensures the $\sigma(\mathbf{v})$ converged to binary values. The activations cannot be quantized using adaptive rounding because they vary with different input data points. Thus, we can only adjust its quantization step size Esser et al. (2020). Denoting the quadratic loss in above equation as $L_q$, the gradients of step size is given by

$$
\frac{\partial L_q}{\partial s} = \begin{cases} \dfrac{\partial L_q}{\partial \hat{\mathbf{x}}} & \text{if } \mathbf{x} > n \\ \dfrac{\partial L_q}{\partial \hat{\mathbf{x}}} \left( \dfrac{\hat{\mathbf{x}}}{s} - \dfrac{\mathbf{x}}{s} \right) & \text{if } 0 \leq \mathbf{x} < \alpha \\ 0 & \text{if } \mathbf{x} \leq 0 \end{cases}, \tag{18}
$$

where all step size in the block will be optimized.

### B.4.2   GENETIC ALGORITHM FOR MIXED PRECISION

---
**Algorithm 2:** Genetic algorithm

---
**Input:** Random initialized population $P_0$ with population size $S$; Iteration $T$, mutation
probability $p$; Hardware performance threshold $\delta$; Hardware measurement function
$H(\cdot)$

$TopK = \varnothing$ ;
**for** *all $t = 1, 2, \ldots, T$-th iteration* **do**
  Evaluate fitness value (Eq. (11)) for each individual ;
  Update and sort $TopK$ based on fitness function;
  **repeat**
      New bitwidth configuration by crossover $\mathbf{c}_{cross} = \text{Crossover}(TopK)$;
      $P_{crossover} := P_{crossover} + \mathbf{c}_{cross}$ if $H(\mathbf{c}_{cross}) < \delta$;
  **until** *Size of $P_{crossover}$ equal to $S/2$*;
  **repeat**
      New bitwidth configuration by mutation $\mathbf{c}_{mutate} = \text{Mutate}(TopK, \text{probability} = p)$;
      $P_{mutate} := P_{mutate} + \mathbf{c}_{mutate}$ if $H(\mathbf{c}_{mutate}) < \delta$;
  **until** *Size of $P_{mutate}$ equal to $S/2$*;
  $P_t = P_{crossover} \cup P_{mutate}$;
  $P_{mutate} = \varnothing, \ P_{crossover} = \varnothing$;
Get the best fitted entry and then do the overall block reconstruction (cf. algorithm 1);
**return** mixed precision model

---

### B.4.3   LATENCY ACQUISITION

We test the latency of quantized neural networks on a self-developed simulator of a precision-variable accelerator for NN. The basic architecture of this accelerator is inspired by typical systolic-matrix multiplication. The accelerator supports the per-channel quantization parameter. The precision of each layer of a NN is highly configurable in this accelerator, supporting 9 types of precision combination: activation: 2-, 4-, 8-bit × weight: 2-, 4-, 8-bit, see Fig. 5a. With the support of scalable function-unit (Sharma et al., 2018), the peak performance of the accelerator is able to achieve corresponding linear improvement as the precision decreases. For example, the peak performance of this accelerator is 256 GMAC/s in 8-bit × 8-bit precision, and it scales to 512 GMAC/s in 8-bit × 4-bit precision and 4 TMAC/s in 2-bit × 2-bit precision. Although this accelerator provides considerable computation resources especially in low precision, the parallelism of the specific layer (like depthwise convolution) and the bandwidth of on-chip buffer is limited. Consequently, actual performance may not scale accurately along with the peak performance, and the final performance differs according to the size and type of layers. The simulator performs cycle-accurate simulation and evaluation for a given NN executed on the accelerator, so we can get an equivalent evaluation by using this simulator. The simulator is available in the provided source codes.

For the acquisition of mobile ARM CPU latency, we adopt the redesigned low-bit GEMM implementation in Han et al. (2020). Fig. 5b shows a brief overview of the low-bit GEMM implementation. Since there is no instruction supporting the bit-width below 8 on ARM architecture, we can

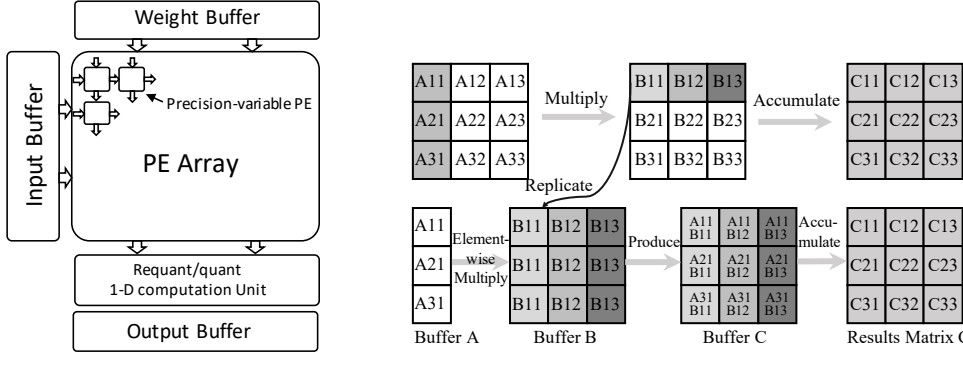

(a) Accelerator design.                    (b) Low-bit GEMM implementation on ARM CPU.

Figure 5: FPGA-based and mobile CPU-based latency acquisition.

not get a higher computation efficiency for extremely low-bit such as 2-bit and 4-bit. But we can acquire a better memory access efficiency. The primary speedup comes from the reduction of data movement. Specifically, we can conduct more times of addition in the same 8-bit register before we have to move it to a 16-bit register to avoid overflow. The lower bit-width is used, the less movement is needed. Together with the optimized data packing and data padding, we can run mixed precision quantization on Raspberry Pi 3B, which has a 1.2 GHz 64-bit quad-core ARM Cortex-A53. Note that this implementation is not optimized for depthwise separable or group convolution, therefore we only verify the latency on ResNets.

### B.4.4 IMPLEMENTATION DETAILS

The ImageNet dataset consists of 1.2M training images and 50K test images. We follows standard pre-process (He et al., 2016) to get 1024 224×224 input images as the calibration dataset. We fold the batch normalization layer into convolution and freeze the BN statistics before post-training quantization. We use Adam optimizer (Kingma & Ba, 2014) to learn the weight rounding and activation range to reconstruct the block output. Note that some layers are not a component of any block, such as the first convolutional layer and the last fully connected layer and the last convolutional layer in the MobileNetV2. These layers use naive layer reconstruction. The batch size of learning is set to 32 and each block will be optimized for $2 \times 10^4$ iterations. The learning rate is set to $10^{-3}$ during the whole learning process. Other hyper-parameters such as the temperature $\beta$ are kept the same with Nagel et al. (2020). For activation step size, we also use Adam optimizer and set the learning rate to 4e-5. Note that we do not implement the gradient scale as introduced in the original paper (Esser et al., 2020). After reconstruction, we will store the sensitivity measured on the calibration dataset. Note that we will store intra-block sensitivity in 2-bit quantization. The sensitivity, as well as hardware performances for each layer, will be stored in a lookup table. When calculating the fitness value and determining the hardware performances in a genetic algorithm, we will check the lookup table. For genetic algorithm, we set the population size to 50 and evolve 100 iterations to obtain the best individual. The first population is initialized by Gaussian distribution and we round the samples to integers in [0, 1, 2], corresponding to bit-width [2, 4, 8]. The mutation probability is set to 0.1. The genetic algorithm usually completes the evolution in only about 3 seconds.

For object detection tasks, we use 256 training images taken from the MS COCO dataset for calibration. The image resolution is set to 800 (max size 1333) for ResNet-18 and ResNet-50, while the image resolution for MobileNetV2 is set to 600 (max size 1000). Note that we only apply block reconstruction in the backbone because other parts of the architecture, such as Feature Pyramid Net, do not have the block structure. Therefore a naive layer reconstruction is applied to the rest of the network. Learning hyper-parameters are kept the same with ImageNet experiments.

