# OpenReview forum: "BRECQ: Pushing the Limit of Post-Training Quantization by Block Reconstruction"
_ICLR.cc/2021/Conference — ICLR 2021 Poster_

### Official Review · AnonReviewer3 · 2020-10-28
**A good paper on post-training inference quantization**

**Rating:** 7
**Confidence:** 4

**Review:**

This paper explores the post-training inference quantization. Based on second-order quantization error analysis, it proposes to reconstruct quantized model in a block level to achieve SOTA accuracy for INT2 weight quantization, which distinguish this paper from previous reported layer-wise reconstruction approach. The proposed approach is intuitive and supported by extensive experiments across a wide range of image classification and object detection tasks.

Overall, this paper is well written. The idea is straightforward but comes from a detailed theoretical analysis and supported by strong experimental results. The authors also proposed solutions to approximate pre-activation Hessian by using Fisher Information Matrix which sounds reasonable and is easy to implement. The authors did comprehensive comparison with SOTA approaches, not just PTQ, but also quantization aware training and mix-precision frameworks.


I have some questions wish to be clarified.

1). It is not clear to me, in the reconstruction process, how does the short-cut (e.g. in ResNet) are handled.

2). In the error analysis, batch normalization layers are not considered. Batch norm will have direct impact on the output activation, hence the quantization bias term. Plus, batch norm statistics will be changed during the reconstruction phase. Could the authors comment on the impact of batch norm layers?

3). The authors use 1024 training samples to do the reconstruction. What happens when more samples are used? What determines the number of samples needed?

4). The data presented seems to focus mainly on weight quantization, while leaving activation in relative higher precision. What happens when quantize activations in 2 bits together with the weights in this approach? What are the challenges?

5). It would have been nice to verify this approach on tasks other than vision, such as speech and NLP tasks.

6). Seems a format error on one of the reference: Zichao Guo, Xiangyu Zhang, Haoyuan Mu, Wen Heng, Zechun Liu, Yichen Wei, and Jian Sun. {SINGLE} {path} {one}-{shot} {neural} {architecture} {search} {with} {uniform} {sampling}, 2020. URL https://openreview.net/forum?id=r1gPoCEKvH.

---

> ### Author Response · Authors · 2020-11-16
> **Response to R3**
>
> Thank you for your positive feedback on this work, we wish we can address your concerns in the response below.
>
> - Q: " It is not clear to me, in the reconstruction process, how does the short-cut (e.g. in ResNet) are handled."
>
>   A: We are sorry we did not make this point clear. During block reconstruction, the block output is the output after the elemental-wise addition of two paths. Therefore, we consider the effect of shortcuts in block reconstruction. In fact, we think the shortcut component in the modern CNN block increases the intra-block dependency, therefore it is more demanding to conduct block reconstruction.
>
> - Q: "In the error analysis, batch normalization layers are not considered. Batch norm will have a direct impact on the output activation, hence the quantization bias term. Plus, batch norm statistics will be changed during the reconstruction phase. Could the authors comment on the impact of batch norm layers?"
>
>   A: For all of our experiments, we fold BN layers into convolutional layers as did in [1], since BN layers require floating-point operations and will slow down the inference. As a result, there is no batch norm layer in the network and no statistics update in the reconstruction phase.
>
> - Q: "The authors use 1024 training samples to do the reconstruction. What happens when more samples are used? What determines the number of samples needed?"
>
>   A: In general, more samples bring higher final performance. Please see our experimental results in Appendix.B.2. In 4-bit quantization, the effect of numbers of samples is trivial. But in 2-bit quantization, more samples can increase 5% accuracy. As for *What determines the number of samples needed?* We think that it depends on the practical situation, that how much data we can get to do PTQ. In this work, we choose 1024 and align experiments setting for all baselines.
>
> - Q:"The data presented seems to focus mainly on weight quantization, while leaving activation in relative higher precision. What happens when quantize activations in 2 bits together with the weights in this approach? What are the challenges?''
>
>   A: In PTQ, 2-bit activation quantization will lead to catastrophic results because activations are quantized at run-time and vary greatly. So we cannot learn a per-element quantization mechanism for activations since they vary with different input images. All we can do in post-training activation quantization is to find a good clipping range.
>
>   The reason why activation quantization can be okay in QAT is that the BN statistics can be updated to remember the activation distribution and adapt to a better mean/variance. In PTQ, it is almost impossible to calibrate accurate BN statistics with limited data, so low-bit activation quantization is harder.
>
> - Q: "It would have been nice to verify this approach on tasks other than vision, such as speech and NLP tasks."
>
>   A: This is a good suggestion for this paper and we would be glad to test BRECQ on other tasks such as NLP. But we did not conduct much research in this area and it takes some time for us to prepare the code and dataset. Considering the limited time for rebuttal, we will try to implement BRECQ on these tasks but we could not promise to complete the experiments.
>
> - Q: "Seems a format error on one of the reference"
>
>   A: Thanks for the advice, we have revised it.
>
>
>
> [1] Raghuraman Krishnamoorthi. Quantizing deep convolutional networks for efﬁcient inference: A whitepaper. arXiv preprint arXiv:1806.08342, 2018.

---

### Official Review · AnonReviewer2 · 2020-10-29
**Paper Not Clear**

**Rating:** 6
**Confidence:** 4

**Review:**

This paper proposes BRECQ which is a new Post Training Quantization (PTQ) method. The goal of the paper is to push the limit of PTQ to low bit precision (INT2). They try to address this by considering both inter and intra-layer sensitivity to find the best update to the model parameters so that the output from a block is minimally changed/perturbed. Furthermore, the authors also consider mixed precision quantization setting.

Empirical results are shown for multiple NNs for image classification and object detection.


While the paper is trying to address an interesting problem and there are a lot of empirical results, but I had a very hard time to follow the paper's main idea and the "final" proposed algorithm. It would be great if the authors could answer the questions below and I will reconsider my score accordingly.


- What is the final algorithm? The data provided in Algorithm 1 is very vague and there are no equations. It is not clear if you are actually using any second-order information since based on Eq 11 it seems that the proposed method is only using the gradient, and not the second order information. If so are the discussion from page 2-4 necessary?

- It is not clear if the equality given in Eq 11 is correct. With the softmax layer the Hessian diagonal would not be the same as gradient^2.


- Why not use the change in loss in Eq 4 (left) instead of using a second-order Taylor series? Second-order Taylor series is an approximation and does not hold for large perturbations. Also when we have access to evaluating the change in loss, why do we even need to focus on measuring the second order term? This does not seem to be correct/necessary at least for uniform quantization.

- An ablation study is actually needed for the above.

- What is the experimental setting for the ablation study in Table 1? while not mentioned, from the accuracy it seems the results are on ImageNet but this is not mentioned. Did the authors consider other bit precision settings?


- In page 3, it is stated that using the Hessian directly is not possible due to memory and computational overhead. However, you can use matrix free methods which do not require forming the Hessian explicitly.

---

> ### Author Response · Authors · 2020-11-16
> **Response to R2 Part II**
>
> + Q: "Why not use the change in loss in Eq 4 (left) instead of using a second-order Taylor series? Second-order Taylor series is an approximation and does not hold for large perturbations. Also when we have access to evaluating the change in loss, why do we even need to focus on measuring the second order term? This does not seem to be correct/necessary at least for uniform quantization. An ablation study is actually needed for the above."
>
>   A: This is an interesting point in PTQ. The reviewer notes it correctly that there exists an alternative training method: QAT with limited images (1024). In fact, we verified this method in our early exploration. The network loss optimization does not perform well, we think the reasons are twofold: (1) There are only limited images in the calibration set, (2) we fold the BN layers before reconstruction to provide higher speed-up in deployment. Without BN to update the activation statistics, the whole network optimization is hard to converge to global minimum.
>
>   In fact, the network reconstruction defined in our paper is very similar to this method. Both of these two methods optimize all network parameters according to the loss computed by the network output. Likewise, it performs worse than block-wise reconstruction.
>
>   The reason why block reconstruction achieves good results might be: (1) most cross-layer dependencies are centered inside a block and thus we can divide the whole network optimization into the block-level, (2) A block with several layers is easy to learn on such a small calibration dataset.
>
>   Here we show the results of the ablation study. (all hyper-parameters are aligned and we update this code in the Google Drive link).
>
>   **Model  |   Precision  |  Block Reconstruction  |  Net Reconstruction   |   Net Loss Optimization**
>
>   Res18	    |  W4A8         | 70.70                                | 68.73                             |  56.20
>
>   Res18	    |  W2A8	   | 66.30                                | 54.15                             |   4.33
>
>   MobV2     |  W4A8         | 71.66                                | 66.68                             |  51.27
>
>   MobV2     |  W2A8         | 59.67                                | 40.76                             |   2.57
>
> + Q: "What is the experimental setting for the ablation study in Table 1? while not mentioned, from the accuracy it seems the results are on ImageNet but this is not mentioned. Did the authors consider other bit precision settings?"
>
>   A: All classification experiments are conducted on ImageNet, sorry that we did not make this clear before. In other bit precision settings, we also observe the same trend in these four kinds of reconstruction granularity. But in general, higher bit-width produces less distinct results because higher bit precision incurs small perturbation and is easy to optimize. Besides, we also provided the implementation for the reviewers to verify the settings that they are interested in.
>
> + Q: "In page 3, it is stated that using the Hessian directly is not possible due to memory and computational overhead. However, you can use matrix free methods which do not require forming the Hessian explicitly"
>
>   A: By mentioning matrix-free optimization, we assume the reviewer is talking about the conjugate gradients method, which indeed avoids the explicit computation of the Hessian. However, the conjugate method only applies to quadratic optimization when weights are not constrained. In this work, weights are quantized and restricted to integers and therefore the conjugate gradient method is not directly applicable. But it would be interesting to see further works about conjugate gradient in quantized neural network optimization. If you have suggestions on other matrix-free methods, we are happy to discuss them.
>
> [1] Markus Nagel, Rana Ali Amjad, Mart van Baalen, Christos Louizos, and Tijmen Blankevoort. Up or down? adaptive rounding for post-training quantization. arXiv preprint arXiv:2004.10568, 2020.
>
> [2] Diederik P Kingma and Jimmy Ba. Adam: A method for stochastic optimization. arXiv preprint arXiv:1412.6980, 2014
>
> [3] Hyeyoung Park, Shun-ichi Amari, and Kenji Fukumizu. Adaptive natural gradient learning algorithms for various stochastic models. Neural Networks, 13(7):755–764, 2000.

---

> > ### Comment · AnonReviewer2 · 2020-11-22
> > **Thanks for Response, but Answers Do Not Address Concerns**
> >
> > I would like to thank the authors for their detailed rebuttal. While I appreciate your response, however, several unresolved points remains. In particular, I find some parts of the paper contradictory and I appreciate it if you could please clarify the points below:
> >
> > 1- Some of the equations and the theorems in the paper are at odds, since the final algorithm is not using Hessian or even the full Fisher Information Matrix, but most of the presented formulas are finally simplified down to gradient^2. For example, one problem is that the paper assumes a gradient of 0, and starts to derive a second order Taylor series, and finally approximates
> > the Hessian with gradient, which should be 0. In fact this is stated in page 2 as an assumption:
> >
> > "Given the pre-trained model is converged to a minimum, the gradients can be safely thought to be close to 0."
> >
> > With a zero gradient then the final metric, which is based on gradient^2 would be zero. (Assuming the calibration dataset comes from the training data, then we will have zero for this metric)
> >
> > Furthermore, given this metric, do we need Theorem 3.1 and its proof?
> >
> >
> > 2- [Repeating this question] "Why not use the change in loss in Eq 4 (left) instead of using a second-order Taylor series? ".
> > The proposed method is basically trying to approximate the change in loss, but you can directly compute the change in loss directly without the need for any Taylor series expansion or any approximation. However, based on the ablation study in the rebuttal, computing the loss directly results in significantly worse results. Assuming those results are correctly obtained, then the only conclusion would be that the Taylor series expansion is not approximate this loss at all, or else it would have achieved very low accuracy as well. If that is the case, then most of the theory and discussions in the paper won't be applicable since it is based on approximating this loss to begin with.
> >
> > I think a more in-depth analysis is needed

---

> > > ### Author Response · Authors · 2020-11-23
> > > **Re: "Thanks for Response, but Answers Do Not Address Concerns"**
> > >
> > > Thank you for your feedback on our rebuttal and we really appreciate your comments on this paper. Please see our response in detail:
> > >
> > > Q: "1- With a zero gradient then the final metric, which is based on gradient^2 would be zero. (Assuming the calibration dataset comes from the training data, then we will have zero for this metric)"
> > >
> > > A: First, we want to clarify that in Page 2 we mention the bar accent denotes the expectation over data points. Therefore, the gradients of weight $\bar{\mathbf{g}}^{\mathbf{w}}=\mathbb{E}[\nabla_{\mathbf{w}}L]\approx 0$ means the expectation of the gradients over data distribution is 0. It cannot guarantee the gradient computed by any batch is 0.
> > >
> > > Second, the **expectation** of gradients and the **expectation of gradients^2** are not the same thing. In page 2 we assume the **expectation** of gradients are close to 0. While in Eq. 7 and Eq. 11 the final metric measures the **expectation of gradients^2**. Apparently, the expectation of squared gradients does not equal to the expectation of squared gradients. $\mathbb{E}[\mathbf{g}]^2\not=\mathbb{E}[\mathbf{g}^2]$.
> > >
> > >
> > >
> > >
> > > ----
> > >
> > >
> > >
> > > Q: "2- Why not use the change in loss in Eq 4 (left) instead of using a second-order Taylor series?. The proposed method is basically trying to approximate the change in loss, but you can directly compute the change in loss directly without the need for any Taylor series expansion or any approximation. However, based on the ablation study in the rebuttal, computing the loss directly results in significantly worse results. Assuming those results are correctly obtained, then the only conclusion would be that the Taylor series expansion is not approximate this loss at all, or else it would have achieved very low accuracy as well. If that is the case, then most of the theory and discussions in the paper won't be applicable since it is based on approximating this loss to begin with."
> > >
> > > A: As we said, this is a very interesting point in post-training quantization. First, we would like to apologize that we misinterpret your question in the pre-rebuttal review.  You mention we can directly measure the loss function, and our previous experiments directly minimize the loss function of the quantized model  $L(\mathbf{w}+\Delta\mathbf{w})$, just like quantization-aware training does. Here is the [link](https://drive.google.com/file/d/1Yd6sHe_zVAoWeuLUDRXxghTjYlCPUG6q/view?usp=sharing) of the output log in these experiments (ResNet-18 W2A8). You can clearly find that **the loss function can be optimized down to 0,** but the test accuracy is only 4%. Apparently, the quantized model **overfits the calibration dataset and does not generalize well across the test set**.
> > >
> > > We rethink your question and find that your suggestion is that we should use the **change in loss**, therefore the objective should be minimizing the discrepancy between $L(\mathbf{w}+\Delta\mathbf{w})$ and $L(\mathbf{w})$. We test this setting and find that it improves the original experiments from 4.33% to 24.54%. However, the result is still poorer than net-wise reconstruction.
> > >
> > > We realize that there might be other reasons for post-training quantization. Typically, the small calibration dataset in PTQ is not enough to represent the whole training dataset, so if we over-emphasize the loss function of the calibration dataset, it might lead to overfitting because the loss function is only a **scalar** and contain less information if there is insufficient data. In comparison, the net-wise reconstruction wants the quantized model to mimic the distribution (all 1k outputs of the network), so it brings more information that relieves the quantized model from overfitting. You can find it in this [link](https://drive.google.com/file/d/1MWes0jAXbPmdF4iSsluVa7bJvzP5NoAq/view?usp=sharing) that the quadratic loss cannot be easily optimized to 0 as CE loss did.
> > >
> > > Finally, let's conclude our findings. "Does Taylor series expansion really approximates this loss?" The answer is yes, but both the accurateness and richness of the approximation in loss contribute to the final performances (maybe the former accounts for underfitting, and the latter accounts for overfitting). Therefore, it is better to mimic the network distribution than the loss value when there is only limited data. From this perspective, we could say BRECQ finds an optimal choice of supervision to guide the quantized model in PTQ where only limited calibration data is available.
> > >
> > > We sincerely thank the reviewer for pointing this out, we did not realize this problem in PTQ at first. Considering the limited time leaving for rebuttal, we cannot revise our manuscripts in one day. But we promise to add the discussion of this part in the final version if the paper can be accepted.

---

> ### Author Response · Authors · 2020-11-16
> **Response to R2 Part I**
>
> We would like to thank you for your thoughtful review and insight on this paper. We have revised our manuscript to make some points clearer. The detailed response is listed below and we hope we have addressed your concerns.
>
> - "Q: What is the final algorithm? The data provided in Algorithm 1 is very vague and there are no equations. "
>
>   A: We define 4 different kinds of reconstruction units, and in our experiments, we find the BLOCK-level is the best choice. The final algorithm is to reconstruct the output block-by-block and use FIM to weigh the importance of activation during reconstruction. Note that our strategy is general and can be compatible with different quantization methods (learning methods).
>
>   Here we choose adaptive rounding for weight quantization and LSQ for activation quantization because these two methods have been proved effective.
>
>   The formulation of the adaptivate rounding is:
>
>   $\hat{w}=s\times\mathrm{clip}(\lfloor \frac{w}{s}\rfloor+\sigma(v),n,p)$, where the variable $v$ can determine rounding-up or rounding-down.
>
>   The formulation of LSQ is:
>
>   $\hat{x}=s\times \mathrm{clip}(\lfloor\frac{x}{s}\rceil,0,p)$, where the quantization step size is learned by STE.
>
>   Now, the revised version adds the reference of equations in the final algorithm. Please take a look at it.
>
> + Q: "It is not clear if you are actually using any second-order information since based on Eq 11 it seems that the proposed method is only using the gradient, and not the second-order information. If so are the discussion from page 2-4 necessary"
>
>   A: First, the diagonal Fisher Information Matrix in Eq.11 acts as a good measure for the second-order information to conduct better quantization. As we mentioned in the paper, prior works that leverage the layer-wise MSE reconstruction only approximate the pre-activation Hessian by $\mathbf{H}^{(\mathbf{z})}=c\times\mathbf{I}$. In our work, we use the diagonal Fisher Information Matrix to approximate the preactivation Hessian and score each output activation with an importance measure. So our approximation is more accurate.
>
>   Second, we understand that there might be some concerns because no real second-order gradients are explicitly computed. But our diagonal FIM approximation overcomes two problems of real second-order gradients: (1) relieve the optimization process from the noises in the Hessian. We recommend the reviewer to refer to a related work, AdaRound[1] and in table 2 they show the real Hessian actually performs worse than the MSE reconstruction. The reason is that, in large-scale deep networks, the curvature information in Hessian is usually noisy. Some works like Adam[2] also prove this point. They approximate the curvature information using only the first-order gradients computation just like ours and achieve good results. (2) reduce the computation overhead and easy to implement. As a PTQ method, such approximation is practical in production.
>
> + Q: "It is not clear if the equality in Eq.(11) is correct. With the softmax layer the Hessian diagonal would not be the same as gradient^2".
>
>   A: Indeed, the softmax layer of the Hessian would not be equal to $g^2$, however, in this section we use FIM to approximate the Hessian. Since the pre-trained model has converged, the outputs of the network are more close to one-hot. So the effect of the softmax layer is smaller and the FIM is very close to Hessian. In Appendx.A.3, we give a detailed discussion of the relationship between Hessian and the FIM.
>
>   See Eq.25 in reference [3] that the diagonal of FIM is equal to $g^2$ with Softmax layer.

---

### Official Review · AnonReviewer1 · 2020-10-29
**Interesting Method and Solid Results**

**Rating:** 8
**Confidence:** 4

**Review:**

Post-training quantization is an important problem, especially for industry. This paper leverages the basic building blocks and conducts a block-wise quantization. Nice results are obtained with the proposed method.

The proposed method is cheap to implement and pushes the post-training quantization to 2-bit.
The measurement problem of mixed precision literature raised in this paper is of insights. This problem may inspire the community to find a better measurement in future work.

Extensive experiments on various methods (handcrafted and designed by NAS), various tasks (classification, detection), various configurations (different bits, latency, model size) are impressive. Various baselines are also included to make the results stronger.


Questions:
The block-diagonal scheme is selected according to experimental results. Is it possible to visualize the real Hessian of stage-wise settings? (it may be impossible to the full Hessian matrix for the whole network). If we see a few non-zero elements at the off-diagonal for the block-wise setting, this choice can be better motivated.

---

> ### Author Response · Authors · 2020-11-16
> **Response to R`1**
>
> Thank you for your insights and feedback on this paper.
>
> Your suggestion about "Is it possible to visualize the real Hessian of stage-wise settings? (it may be impossible to the full Hessian matrix for the whole network)" is of great value. We visualize a Hessian matrix on a tiny ResNet. The code can be found in this Google Colab Link: https://colab.research.google.com/drive/1hLe-Avgtdy0l5gKibR_zr2SjqApang9D?usp=sharing.
>
> The tiny ResNet contains 3 stages with channels of {4, 8, 16} (so that we can calculate the Hessian easily.) The ResNet has 99.37\% accuracy on the MNIST dataset. We compute the Hessian of the first stage. The first stage has 2 blocks, where each block has 2 layers. Each layer in the stage has 144 number of elements.
> We can find that while most of the values in the Hessian matrix are close to 0, the upper left Hessian (which corresponds to the first block) indeed has slightly higher absolute values, which confirms the observation in our paper. Whereas most of the inter-block Hessian, which is the upper right and lower left part of the Hessian, is close to 0.

---

### Official Review · AnonReviewer4 · 2020-11-02
**Couldn't understand paper, results seem strong**

**Rating:** 7
**Confidence:** 1

**Review:**

I couldn't follow the method described in the paper. The authors are basically trying to address post-training quantization by perturbing the the weights of a trained DNN. The goal is to perturb the weights so that the quantized DNN will behave similar to the original full-precision DNN. The authors draw a link between this optimization problem and optimizing for the "reconstruction" of the output activations of a block (see Equation 7). The technique BRECQ, shown in Algorithm 1, is basically to optimize the perturbation of the weights for the right hand side of Equation 7 for each block of a DNN.

The method only seems to work on ReLU networks, so it's restricted to CNNs, but this is still extremely useful.

The experimental results are very strong. BRECQ is the first to achieve 4-bit integer post-training weight quantization that nearly matches the fp32 baseline on ImageNet. Even at 2-bit, BRECQ can often get within 5% of the baseline while other methods leave the model with inoperable loss of accuracy.

I couldn't pinpoint any major flaws in the paper, and the results are extremely impressive.

---

> ### Author Response · Authors · 2020-11-16
> **Response to R4**
>
> Thanks for your reviews and feedback on this paper. Here is the detailed response.
>
> + Q: "I couldn't follow the method described in the paper. The authors are basically trying to address post-training quantization by perturbing the the weights of a trained DNN. The goal is to perturb the weights so that the quantized DNN will behave similar to the original full-precision DNN. The authors draw a link between this optimization problem and optimizing for the "reconstruction" of the output activations of a block (see Equation 7). The technique BRECQ, shown in Algorithm 1, is basically to optimize the perturbation of the weights for the right hand side of Equation 7 for each block of a DNN."
>
>   A: Your summary of this paper is right, and we obtain the final results through some theoretical analysis. Here is the core analysis process of this paper:
>
>   1. Based on Eq.7, we show that the second-order error can be transformed into the output (pre-activations). Reconstructing the network output will consider all the cross-layer dependency.
>   2. Based on Eq.9, we show that using more layers to conduct reconstruction would lead to higher approximation error for quantization bias.
>   3. To achieve the best trade-off between the cross-layer dependency and first-order approximation error, we choose an intermediate level: BLOCK, to perform reconstruction. And we find this rule applies to many architectures and tasks.
>
> + Q: "The method only seems to work on ReLU networks, so it's restricted to CNNs, but this is still extremely useful."
>
>   A: Theorem 1 actually can be applied to any activation function that has first-order gradients, since the approximation is made by $\Delta z\approx J\Delta w$. But Theorem 2 is indeed built upon ReLU network, so there are indeed some restrictions. But as you say, ReLU networks are popular nowadays for many tasks besides CV, such as the NLP tasks. For example, the FFN in Transfomer is formed by FC and ReLU. Thanks for pointing this out, and we will try to analyze and evaluate the effect of BRECQ on other types of models in the future.

---

### Author Response · Authors · 2020-11-16
**General Response**

We feel grateful for the reviewers' positive feedback towards BRECQ. Some of them raise the concern that the method is not clear. We would like to make some rough clarifications about this point.

BRECQ studies the loss degradation in post-training quantization by approximating the Taylor expansion and Gauss-Newton approximation of the Hessian matrix. We show that the cross-layer dependence can be computed by measuring the distance of the output of the whole network. However, if more layers are considered, the distance of output will become an inaccurate signal and prohibits the optimization. To reach the best tradeoff between them, we use the building block as the basic reconstruction unit. This method requires no hyper-parameters and is simple to perform. We conduct extensive experiments on various models and tasks to verify the algorithm.

We also update our manuscripts, the change we made includes:
1. In Section 3.2, we add a numbered list to introduce the 4 different kinds of reconstruction granularity. It is easier to compare the difference between these 4 levels.
2. In Algorithm 1, we add a reference to the learning strategies.
3. We put related works in Section 5 to make the main text more self-contained.

The detailed explanation sees the response to each reviewer.

---

### Comment · ~Itay_Hubara6 · 2021-01-18
**Several issues with the code**

Hi,

Very interesting paper. I tried running your code to reproduce the results and I encountered several issues. Please let me know if you can help with this:

1. No module named 'mc' - over the web I found another repo (https://github.com/youjiangxu/seqvlad-pytorch/issues/2) that had the same issue and thus tried to change the read_from argument in BaseDataset class to 'fs'. Am I right to do so?
2. Next I received the following error: "import linklink.dali failed, linklink version should >= 0.2.0 " I can't find an open-source linklink package.
3. Finally, I tried to take the essence of the paper and just apply it in my own environment and failed to achieve the reported accuracy boost - any idea what is missing?

---

> ### Author Response · Authors · 2021-01-20
> **Thanks for reporting this**
>
> Hi,
>
> Thank you for your interest in this work, we are cleaning the code and updating some experiments right now, we will include the public runnable code in the camera-ready version.
>
> As for your own environment, which bit-width did you choose?

---

> > ### Comment · ~Itay_Hubara6 · 2021-01-24
> > **Re:Several issues with the code**
> >
> > Thank you for your reply. I experimented with both 2-bit and 4-bit (activation and weights) - does it matter?
> > I mainly investigate the layer-wise reconstruction

---

### Decision · Program_Chairs · 2021-01-07
**Final Decision**

**Decision:**

Accept (Poster)

**Comment:**

This paper proposes a new method for post-training quantization, achieving very good results. After the author's response, all the reviewers were positive. There were some issues regarding clarity, and about explaining why the methods work better than just optimizing the loss, but I think the reviewers were eventually satisfied.  Following some info after the author's response phase, I'll just ask the authors to verify their published code works with publicly available PyTorch packages, so their method could be easily used.